# Interfacial engineering of metal-insulator-semiconductor junctions for efficient and stable photoelectrochemical water oxidation

Ibadillah A. Digdaya[1], Gede W.P. Adhyaksa[2], Bartek J. Trześniewski[1], Erik C. Garnett[2] & Wilson A. Smith[1]

Solar-assisted water splitting can potentially provide an efficient route for large-scale renewable energy conversion and storage. It is essential for such a system to provide a sufficiently high photocurrent and photovoltage to drive the water oxidation reaction. Here we demonstrate a photoanode that is capable of achieving a high photovoltage by engineering the interfacial energetics of metal–insulator–semiconductor junctions. We evaluate the importance of using two metals to decouple the functionalities for a Schottky contact and a highly efficient catalyst. We also illustrate the improvement of the photovoltage upon incidental oxidation of the metallic surface layer in KOH solution. Additionally, we analyse the role of the thin insulating layer to the pinning and depinning of Fermi level that is responsible to the resulting photovoltage. Finally, we report the advantage of using dual metal overlayers as a simple protection route for highly efficient metal–insulator–semiconductor photoanodes by showing over 200 h of operational stability.

[1] Materials for Energy Conversion and Storage (MECS), Department of Chemical Engineering, Delft University of Technology, Van der Maasweg 9, 2629 HZ Delft, The Netherlands. [2] Center for Nanophotonics, AMOLF, Science Park 104, 1098 XG Amsterdam, The Netherlands. Correspondence and requests for materials should be addressed to W.A.S. (email: w.smith@tudelft.nl).

Photoelectrochemical (PEC) water splitting has been envisioned as a sustainable approach to produce clean and renewable fuel by the direct conversion of solar to chemical energy[1–3]. One important step during the PEC process is charge separation, which is determined by the energetics of the semiconductor/liquid junction, or by a buried rectifying junction[4]. The buried junction can be established by variations in doping (homojunction), offsets in the conduction and valence band levels at an interface (heterojunction), or by differences in Fermi level at a metal-semiconductor (MS) interface (Schottky junction)[5–10]. For example, when an n-type semiconductor is in contact with a high work function metal, the electron Fermi-level falls close to the valence band at the interface and creates a potential barrier that repels the majority carriers and selectively conducts the minority carriers[11]. The height of this potential barrier determines the built-in voltage, and therefore the maximum achievable photovoltage. However, direct contact between the semiconductor and metal can lead to surface recombination due to the presence of metal-induced gap states that can cause Fermi level pinning[12–14], thus diminishing the barrier height of the Schottky MS junction. This problem can be effectively tackled by inserting a thin insulator layer that separates the metal and semiconductor, known as a metal–insulator–semiconductor (MIS) junction.

The MIS junction concept has recently attracted considerable attention both in photovoltaic (PV) and PEC applications, mainly due to the simplicity of device fabrication and processing that only requires a thin passivating tunnel dielectric and a metal, as well as the versatility of obtaining the desired barrier height by the wide selection of metals with various work functions[15,16]. The major challenge of MIS structures for solar energy conversion devices, however, is that the metal contact should completely cover semiconductor surface in order to have a homogenous formation of the Schottky junction[17]. This required metal coverage can lead to a significant increase of light reflection, reducing the light absorption in the semiconductor, and consequently decreasing the photocurrent output[17]. Despite this unfavourable structure, advanced nanoscale-structuring of metal contacts have been successful to partially circumvent the optical problem by allowing light to transmit to the absorber layer through the contactless surface while still enabling charge collection through nanostructured metal contacts[18–21].

The key advantage of MIS structures for PEC devices is the fact that metals have a high density of states for efficient collection and transport of minority carriers. This means that any collected minority charge carriers can be readily transferred to the electrolyte through the catalyst to drive an electrochemical reaction instead of flowing through the external circuit as in PV cells. This simple route allows the ability to use thin metals (that is, 1–10 nm) as the charge collector as well as the carrier conduction mediator[22–25]. The thin metal film allows the reaction to occur throughout the whole surface, and thus enabling the photogenerated current to flow perpendicular to the metal film. Additionally, the complete coverage of the metal film results in a buried Schottky junction and allows the Fermi level of the metal to float and adjust with the current density and kinetics of the reaction. Furthermore, a thin metal contact allows for a significant portion of light to transmit into the absorber layer for photocurrent generation.

Despite the great advantage of the MIS structure for solar water splitting, there remains a major trade-off between the high efficiency and the long-term durability. Therefore, many efforts have also concentrated on protecting the photoelectrodes either using a stable oxide insertion layer[11–13] or corrosion-resistant overlayers[26–28]. Another protection scheme can be achieved by tuning the metallic component of the MIS structure. Nickel (Ni)

is an attractive material that has all the functionalities required for MIS photoanodes: high work function for high photovoltage generation, high catalytic activity for water oxidation and high chemical stability in strongly alkaline solutions[29,30]. Kenney et al. have shown that a thin Ni film (2 nm) was sufficient to create a Schottky junction with the semiconductor and readily oxidized to $NiO_x$, forming a Ni/$NiO_x$/electrolyte interface which results in a higher effective work function of the composite layers[27]. However, such a thin Ni layer was not able to completely protect the underlying photoanode in a highly corrosive electrolyte at pH 14. As a result, the performance of the device started to degrade after 24 h of operation in 1 M KOH electrolyte solution. On the other hand, it was also found that the thicker Ni (that is, 5 nm) was able to avoid corrosion in 1 M KOH, but suffered from low photovoltage due to the low Schottky barrier height formed by Si/$SiO_x$/Ni[27].

Herein we demonstrate a MIS photoanode that can yield a high efficiency and high stability by engineering the metal–insulator and insulator–semiconductor interfaces. Specifically, we introduce a high quality tunnelling $Al_2O_3$ layer that is capable to substantially reduce the Fermi-level pinning and allows for Schottky junction formation between the metal and semiconductor. Additionally, we use two metals to decouple the metal properties for Schottky junction formation and to protect and promote catalytic functionalities for water oxidation. We investigate the role of the bi-metallic layers to the energetics at the rectifying junction that is responsible for the high photovoltage of the photoanode. Finally, using a simple yet effective protection strategy, we demonstrate a more than 200-h operation of an MIS photoanode that shows constant high photocurrents in a strong base solution.

## Results

**Fabrication of photoanodes.** MIS photoanodes were fabricated using moderately phosphorus-doped (100) n-type silicon (Si) wafers (resistivity 0.1–0.3 $\Omega$ cm, thickness 525 $\mu$m). The Si wafers were first cleaned using Piranha solution containing a 3:1 mixture of $H_2SO_4$ and $H_2O_2$ to remove particles and organics. Next, the Si wafers were dipped in HF (2%) for 2 min to etch the native oxide, followed by rinsing using deionized water and drying using $N_2$ gas. The interfacial silicon oxide layer were grown on the Si substrate in a Radio Corporation of America Standard Clean (RCA-SC2) solution (from hereafter called $SiO_{x,RCA}$, unless specified otherwise), containing a mixture of $H_2O$, HCl and $H_2O_2$ with a volume ratio of 5:1:1 at 75 °C for 10 min, resulting an oxide thickness of 1.8 nm, as measured by ellipsometer. The dielectric layer was immediately deposited onto the Si by atomic layer deposition (ALD) of approximately 1-nm thick aluminium oxide ($Al_2O_3$). The Schottky contact was formed by depositing 2 nm of platinum (Pt) by radio frequency (rf) sputtering. Finally, the photoanode was completed by depositing 4 nm of Ni that acts both as a catalyst and a protection layer. The metallic components (that is, Pt and Ni) were kept very thin to minimize the light reflection while still providing sufficiently high Schottky barrier for high photovoltage generation and effective protection for the underlying layers. The schematic structure of the MIS photo-anode is shown in Fig. 1.

**PEC performance.** The PEC performance of the MIS photoanodes was evaluated by performing cyclic voltammetry (CV) in 1 M potassium hydroxide (KOH) under simulated solar illumination in three electrode measurements without correction for resistance losses in the electrolyte solution. Figure 2 shows the evolution of current-potential ($j$–$V$) characteristics of the photoanodes over the course of 18 h. The initial photocurrent onset potential (defined as the potential required to achieve an

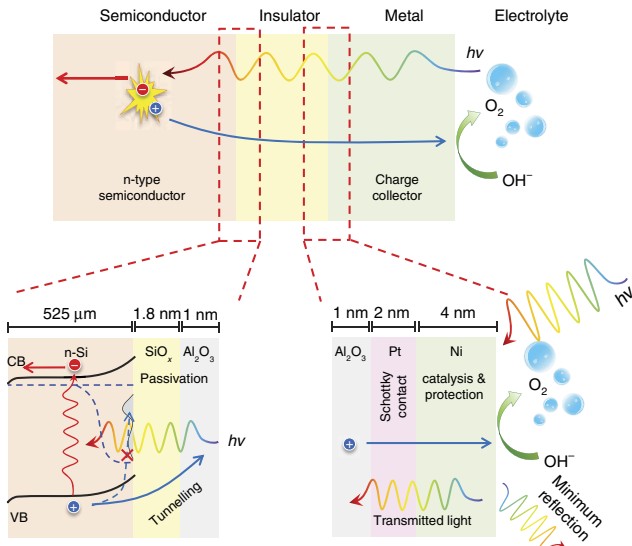

**Figure 1 | Metal–insulator–semiconductor photoanode.** Schematic of planar MIS photoanodes for water oxidation and magnification of the interfaces, showing functionalities of each layer.

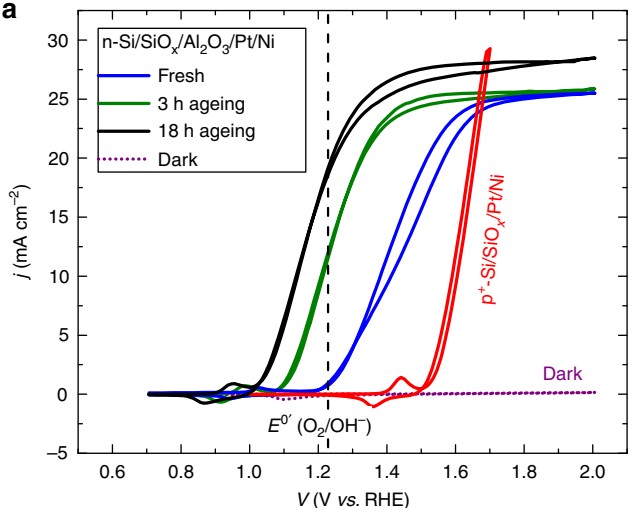

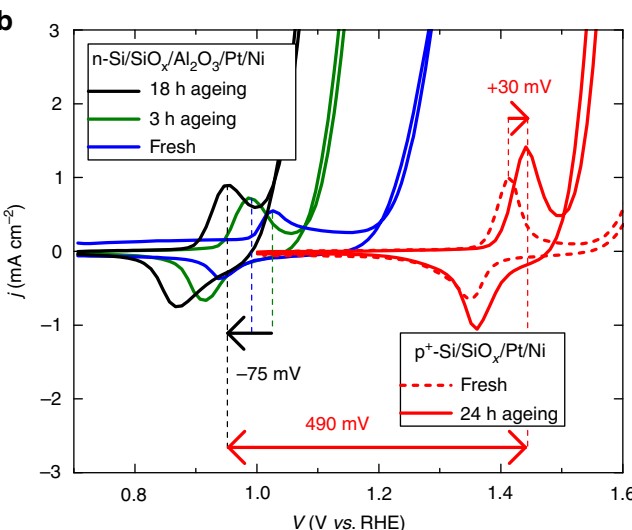

**Figure 2 | Cyclic voltammetries. (a)** Three-electrode (photo) current density versus applied voltage (j–V) curves of n-Si/SiO$_{x,RCA}$/Al$_2$O$_3$/Pt/Ni photoanode during ageing in 1 M KOH under simulated AM1.5 illumination. A total of three illuminated voltammetry scans are shown: one for initial fresh sample (blue line), one after 3 h (green line) and one after 18 h (black line) of ageing in the electrolyte. The dark voltammetry scan of the n-Si/SiO$_{x,RCA}$/Al$_2$O$_3$/Pt/Ni is shown as the dotted purple line. For comparison, the j–V behaviour of the non-photoactive p$^+$-Si/SiO$_{x,RCA}$/Pt/Ni is shown (red line). The black vertical dashed line indicates the potential for water oxidation, $E^{0\prime}$ (O$_2$/OH$^-$). **(b)** The magnification of Fig. 2a, showing the shift of the redox peaks of Ni over the course of 18 h.

anodic current of 100 µA cm$^{-2}$) was − 56 mV, and cathodically shifted to − 168 mV relative to the formal potential for water oxidation ($E^{0\prime}$ (O$_2$/OH$^-$) = 1.23 V versus a reversible hydrogen electrode, RHE) after 3 h in the 1 M KOH electrolyte. The photocurrent onset potential further shifted to − 233 mV relative to $E^{0\prime}$ (O$_2$/OH$^-$), and the photocurrent density of the photoanode was 19.2 mA cm$^{-2}$ at $E^{0\prime}$ (O$_2$/OH$^-$). The series resistance of the system was 4.24 Ω cm$^2$, as measured using electrochemical impedance spectroscopy (EIS), and the photocurrent density at $E^{0\prime}$ (O$_2$/OH$^-$) after compensation for the series resistance was 25 mA cm$^{-2}$ (see Supplementary Fig. 1). The dark current was measured to be close to zero, implying that the high observed current under illumination was related to the photogeneration of charge carriers. The saturated photocurrent density of the photoanode was increased from 25.5 mA cm$^{-2}$ during the first scan to 28.5 mA cm$^{-2}$ after 18 h in contact with 1 M KOH electrolyte (hereafter we define a prolonged contact with KOH electrolyte in the absence of an applied potential as the ageing process[31]). The increase of the limiting photocurrent density is attributed to the oxidation of the thin Ni (4 nm) to the more transparent NiO$_x$ or Ni(OH)$_2$ during the ageing process that reduced the light reflection (see Supplementary Fig. 2), and thus allowing for more light transmission to the absorber layer. The photovoltage of this structure was estimated by comparing the onset potential of the photoanode under illumination and the non-photoactive p$^+$-Si/SiO$_{x,RCA}$/Pt/Ni (here the degenerate p$^+$-Si simply acted as a conductive substrate), and was measured to be 490 mV. The equivalent PV response[21,32] indicated that the equivalent open-circuit voltage ($V_{oc}$) was 496 mV (see Supplementary Note 1 and Supplementary Fig. 3).

Figure 2b shows the magnification of Fig. 2a near the onset potential for water oxidation. The initial oxidation peak of Ni within the n-Si/SiO$_{x,RCA}$/Al$_2$O$_3$/Pt/Ni photoanode was − 390 mV relative to the initial Ni oxidation peak in the non-photoactive p$^+$-Si/SiO$_{x,RCA}$/Pt/Ni. This value suggests that the initial photovoltage of the photoanode during the first scan was 385 mV. After 18 h in contact with KOH, the redox peaks of the non-photoactive p$^+$-Si/SiO$_{x,RCA}$/Pt/Ni anodically shifted by + 30 mV. This potential shift is attributed to the incorporation of Fe into the Ni(OH)$_2$ on the Ni surface during ageing process in a non-purified KOH electrolyte[31,33], which is responsible for the catalytic activation of

Ni for the oxygen evolution reaction. In contrast, the redox peaks of Ni within the n-Si/SiO$_{x,RCA}$/Al$_2$O$_3$/Pt/Ni photoanode shifted cathodically by − 75 mV relative to the initial scan. This cathodic shift of the Ni redox peaks in the photoanode indicated that the photovoltage of the photoanode indeed increased during the ageing process in KOH. The increase of the photovoltage of the photoanode was further ascertained by comparing the electrochemical open-circuit potential (OCP) of the photoanode measured against a reference electrode (scaled in RHE) in the dark and under illumination (Fig. 3). The change in the OCP between the dark and illuminated conditions indicates the photovoltage of the photoanode. Consistently, the OCP of the sample after the first scan was 400 mV, and increased to 490 mV after 18 h in the electrolyte, showing excellent agreement with the j–V measurements shown in Fig. 2.

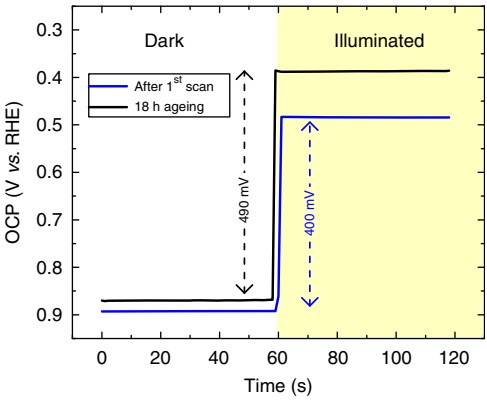

**Figure 3 | Electrochemical open-circuit potential measurement.**
Open-circuit potential versus the reference electrode (scaled in RHE) of the
n-Si/SiO$_{x,RCA}$/Al$_2$O$_3$/Pt/Ni photoanode after the first voltammetry scan
and after 18 h ageing in 1 M KOH measured in the dark and under
illumination. The change in OCP between the dark and illuminated
conditions indicates the photovoltage of the photoanode.

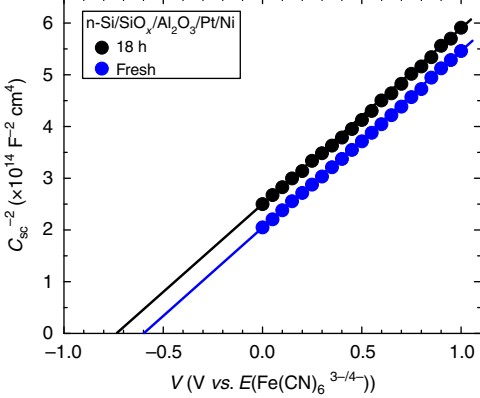

**Figure 4 | Mott–Schottky relationship.** Mott–Schottky plots ($C_{sc}^{-2}$–$V$) of
the inverse square space charge capacitance as a function of applied voltage
relative to the redox potential of Fe(CN)$_6^{3-/4-}$ for the n-Si/SiO$_{x,RCA}$/Al$_2$O$_3$/
Pt/Ni photoanode before (blue) and after 18 h ageing (black).

The energetics of the semiconductor at the Schottky junction
were evaluated by using EIS in a solution containing an
electrochemically reversible one-electron Fe(CN)$_6^{3-/4-}$ redox
couple[9,10]. The space-charge capacitances ($C_{sc}$) of the
semiconductor were obtained by fitting the EIS results with an
equivalent electronic circuit that consisted of parallel resistor and
parallel capacitor, corresponding to the space-charge region of the
semiconductor (Supplementary Fig. 4). The flat band potential
($E_{fb}$) was estimated by taking the value of the intercept between the
extrapolated linear region of the inverse square space-charge
capacitance ($C_{sc}^{-2}$) with the x-axis in the Mott–Schottky plot in
Fig. 4. The $E_{fb}$ of the fresh n-Si/SiO$_{x,RCA}$/Al$_2$O$_3$/Pt/Ni was $-0.6$ V
versus Fe(CN)$_6^{3-/4-}$ and the aged sample was $-0.73$ V
versus Fe(CN)$_6^{3-/4-}$. The slope of the linear region of the $C_{sc}^{-2}$
of both samples was $3.35 \pm 0.02 \times 10^{14}$ F$^{-2}$ cm$^4$ V$^{-1}$, correspond-
ing to a donor density ($N_D$) of $3.54 \times 10^{16}$ cm$^{-3}$ (see Supplemen-
tary Note 2), which also implies a corresponding resistivity ($\rho$) of
$\sim 0.185 \, \Omega$ cm, in accordance with the range specified by the
manufacturer of the Si wafer (0.1–0.3 $\Omega$ cm). The calculated barrier
height of the Si within the fresh and the aged n-Si/SiO$_{x,RCA}$/Al$_2$O$_3$/
Pt/Ni photoanode was 0.77 and 0.9 eV, respectively
(Supplementary Note 2). Such a large barrier height should give
rise to a strong inversion layer near the Si surface, and thus would
result in a large observed photovoltage, mainly due to the
associated increase of band bending, and the subsequent
improvement of charge-carrier extraction as well as separation
inside the semiconductor.

To confirm the composition of the n-Si/SiO$_{x,RCA}$/Al$_2$O$_3$/Pt/Ni
after the ageing treatment, X-ray photoelectron spectroscopy (XPS)
experiments were performed using low energy ion etching with an
average etching rate of approximately 3 Å per step. Figure 5a shows
the Ni 2p XPS peaks on the sample starting from step 0 (no
etching) to step 10 (approximate depth of 3 nm). The results
indicated that in an alkaline solution the Ni surface mostly
transformed to Ni(OH)$_2$, and prolonged ageing partially modified
the bulk Ni into NiO$_x$, as observed after three steps of etching and
reached a maximum after five steps. Strong Ni 2p signals
corresponding to the metallic Ni were observed and became
dominant as the etching depth increased to ten steps. These XPS
spectra substantiate that the initial metallic Ni film (thickness of
4 nm) was transformed into a Ni/NiO$_x$/Ni(OH$_2$) composite after
18 h in contact with aqueous KOH solution. Figure 5b shows the
XPS depth profiling analysis corresponding to Ni 2p, Pt 4f, Al 2s
and Si 2p. The Al 2s signal was used in this analysis due to the

difficulty to identify the Al 2p signals that overlapped with the
Pt 4f (Supplementary Note 3). In total, 36 etching steps were taken
and the full elemental scan can be found in Supplementary Fig. 6.
Consistently with the previous study, oxidation of the composite
film by the electrolyte did not go further than $\sim 2.5$ nm (ref. 27), as
indicated by the dominant signal of metallic Ni at etching step 10
(approximate depth 3 nm), as well as the absence of NiO or
Ni(OH)$_2$ signals at the Pt interface. This implies that the Al$_2$O$_3$ and
Si underneath were still fully protected by the Ni after 18 h.
Without Ni and Pt, Al$_2$O$_3$ easily corroded in 1 M KOH, as
indicated by the absence of Al 2p signal in Supplementary Fig. 7.

The improvement of the barrier height of the n-Si/SiO$_{x,RCA}$/
Al$_2$O$_3$/Pt/Ni photoanode after 18 h of ageing in KOH solution
was closely related to the inability of the 2 nm Pt to completely
screen the charge at the Si interface, and thus the effective
screening was also influenced by the increased work function of
the Ni surface layer upon oxidation in a strongly alkaline
electrolyte. Previous studies have shown that the effective metal
screening and the associated barrier height in a bimetal Schottky
structure depends on the inner metal thickness[20,34]. The XPS
depth profiling analysis equipped with ion etching in Fig. 5 shows
that the Ni film was partially transformed into Ni(OH)$_2$ and NiO$_x$
on the surface after 18 h in 1 M KOH. According to the Pourbaix
diagram (Supplementary Fig. 8), Ni readily oxidizes to NiO$_x$ even
in a slightly alkaline solution (pH > 7), and therefore prolonged
contact with a pH 14 solution might accelerate and exacerbate the
oxidation process of Ni. The NiO$_x$ and Ni(OH)$_2$ phases are
known to have a large work function in the range of 5.2–5.6 eV
(refs 35,36), close to the vacuum work function of Pt. Although Pt
has a sufficiently high work function to produce a large Schottky
barrier, its effective work function may differ from the ideal value,
and is usually lower when in contact with a dielectric material
than when in vacuum. The decrease of the metal work function is
essentially due to the existence of intrinsic states on the surface of
the dielectric layer whose energy level is at the so-called charge
neutrality level, which is the location of the highest occupied state
in the dielectric band gap[37,38]. These intrinsic states tend to pin
the metal Fermi level and lower the effective work function of the
metal relative to its vacuum value. The magnitude of the Fermi
level shift depends on the pinning strength of the dielectric layer,
that is, the pinning factor that ranged from 0 for perfect pinning
and 1 for no pinning. Previous study has shown that Al$_2$O$_3$ has
pinning factors between 0.63 to 0.69 (ref. 37), and can effectively
lower the work function of Pt (vacuum work function of 5.6 eV)

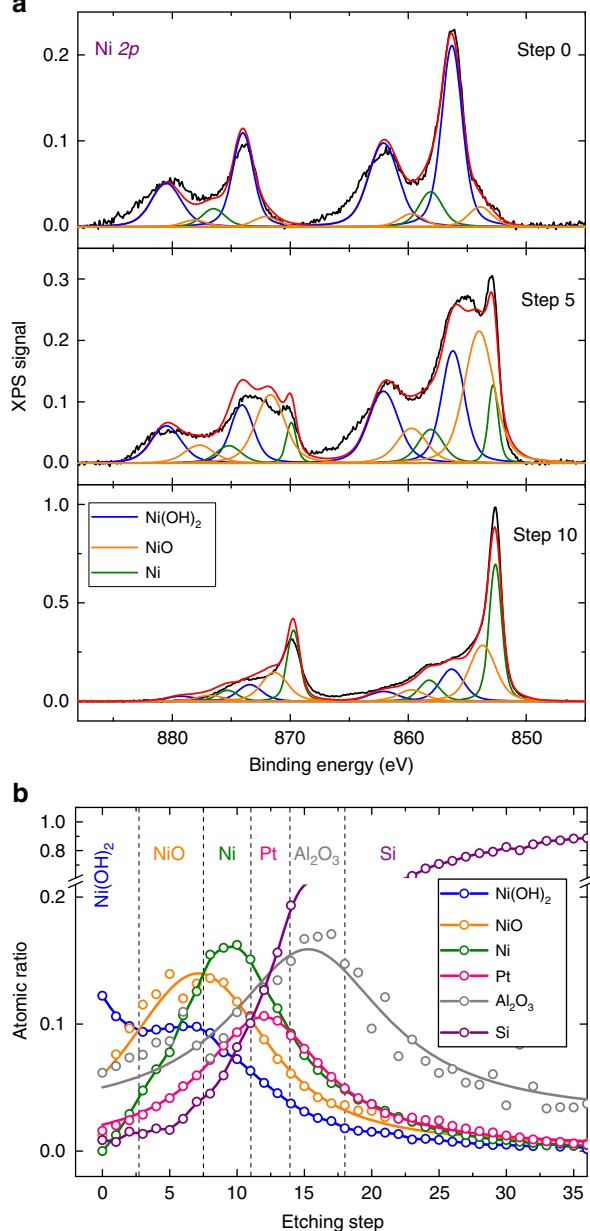

**Figure 5 | X-ray photoelectron spectroscopy spectra of n-Si/SiO$_{x,RCA}$/Al$_2$O$_3$/Pt/Ni after 18 h ageing.** (**a**) Ni 2$p$ signals as a function of etching step; step 0 (no etching), step 5 and step 10 (approximate depth of 3 nm), showing variation of Ni phases. (**b**) Depth profiling analysis of the composite system. The XPS signals corresponded to Ni 2$p$, Pt 4$f$, Al 2$s$ and Si 2$p$. The Al 2$s$ signal was used in this analysis due to the overlapping peak locations of the Al 2$p$ and the Pt 4$f$. The sample was etched using ion beam with an average etching rate of approximately 3 Å per step.

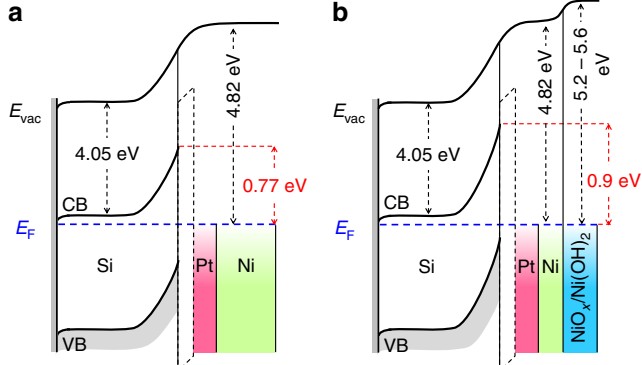

**Figure 6 | Energy band diagrams.** Representative energy band diagrams of the n-Si/SiO$_{x,RCA}$/Al$_2$O$_3$/Pt/Ni before (**a**) and after prolonged ageing of 18 h in KOH solution (**b**). After 18 h, the metallic Ni film was partially transformed into NiO$_x$ and Ni(OH)$_2$. The increase of barrier height of the aged sample was because the increase of work function of Ni upon oxidation to NiO$_x$ and Ni(OH)$_2$.

ageing treatment in aqueous KOH solution, shown in Fig. 7a,b. The vacuum work function of Ni is known to vary from 5.22 to 5.35 eV depending on the crystal facet, and the vacuum work function of Pt is 5.64 eV for polycrystalline films. After 18 h, the onset potential of the n-Si/SiO$_{x,RCA}$/Al$_2$O$_3$/Ni was $-160$ mV relative to $E^{0'}$ (O$_2$/OH$^-$), and the photovoltage was 410 mV (see Supplementary Fig. 9 for the initial $j$–$V$ curve of the n-Si/SiO$_{x,RCA}$/Al$_2$O$_3$/Ni). The initial $E_{fb}$ of the n-Si/SiO$_{x,RCA}$/Al$_2$O$_3$/Ni was $-0.5$ V versus Fe(CN)$_6^{3-/4-}$ (Supplementary Fig. 10) and increased to $-0.6$ V versus Fe(CN)$_6^{3-/4-}$ after 18 h of ageing. Considering the Ni surface layer already oxidized to NiO$_x$ after 18 h in 1 M KOH, and the Ni did not oxidize further than 2.5 nm, the lower photovoltage and $E_{fb}$ observed in n-Si/SiO$_{x,RCA}$/Al$_2$O$_3$/Ni/NiO$_x$ than the n-Si/SiO$_{x,RCA}$/Al$_2$O$_3$/Pt/Ni/NiO$_x$ suggests that the inner metal still partially screened the charge and its work function contributed substantially to the effective barrier height in a multilayer Schottky structure.

Figure 7a also compares the $j$–$V$ characteristics of the samples with and without the Al$_2$O$_3$ layer. Without Al$_2$O$_3$, the onset potentials were $-60$ and $+22$ mV relative to $E^{0'}$ (O$_2$/OH$^-$) and the photovoltages were only 310 and 230 mV for samples with the SiO$_{x,RCA}$ and the SiO$_{x,native}$, respectively. Mott–Schottky plots in Fig. 7c indicate that the $E_{fb}$ was only $-0.56$ V versus Fe(CN)$_6^{3-/4-}$ for the n-Si/SiO$_{x,RCA}$/Pt/Ni and $-0.46$ V versus Fe(CN)$_6^{3-/4-}$ for the n-Si/SiO$_{x,native}$/Pt/Ni, and there were no substantial changes between the initial conditions and after 18 h ageing. The constant $E_{fb}$ and the low photovoltage of the n-Si/SiO$_x$/Pt/Ni photoanode, despite the high work function of the surface layers, is attributable to the strong pinning of the Fermi level at the Si/SiO$_x$ interface. The Fermi level pinning was even worse when the SiO$_{x,native}$ was used in place of SiO$_{x,RCA}$, as observed by the lower photovoltage and barrier height of the n-Si/SiO$_{x,native}$/Pt/Ni in Fig. 7a,c, respectively. Fundamentally, Fermi level pinning arises in response to incomplete termination of the Si surface, resulting in an asymmetric bonding situation, that is, dangling bonds, that introduces energy states inside the surface band gap[39,40]. These energy states tend to pin the Fermi level of the semiconductor, and dictates the barrier height by their energy levels instead of by the metal work function. On the other hand, Al$_2$O$_3$ is known for its high level of chemical passivation that is capable of saturating dangling bonds on the Si surface, and thus has been used extensively for surface passivation of Si[41,42]. Therefore, we attribute the high observed photovoltage of the n-Si/SiO$_{x,RCA}$/Al$_2$O$_3$/Pt/Ni MIS structure to the high quality of surface chemical

down to 5.1 eV. By using the Schottky rule, the $E_{fb}$ of $-0.6$ V versus Fe(CN)$_6^{3-/4-}$ of the fresh sample implies a corresponding barrier height of 0.77 eV and an effective work function of Pt/Ni metal bilayers of 4.82 eV on the n-Si/SiO$_{x,RCA}$/Al$_2$O$_3$. On the other hand, the oxidized NiO$_x$ on Pt within the aged sample was isolated from Al$_2$O$_3$, and thus was not influenced by the pinning of the Al$_2$O$_3$ and would likely to maintain its high work function for Schottky junction formation with Si, as illustrated in Fig. 6.

The effect of the interfacial metallic layer was analysed by comparing $j$–$V$ curves and Mott–Schottky plots of the MIS devices with Pt (2 nm)/Ni (4 nm) and Ni (6 nm) after 18 h of

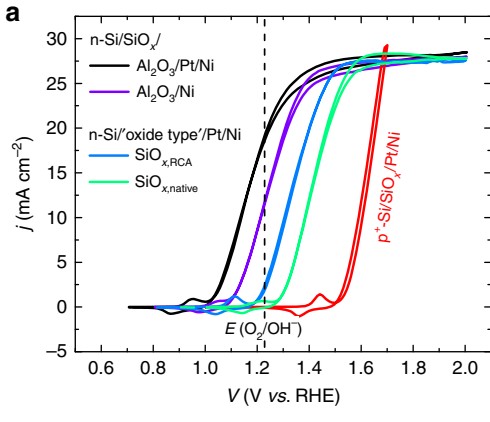

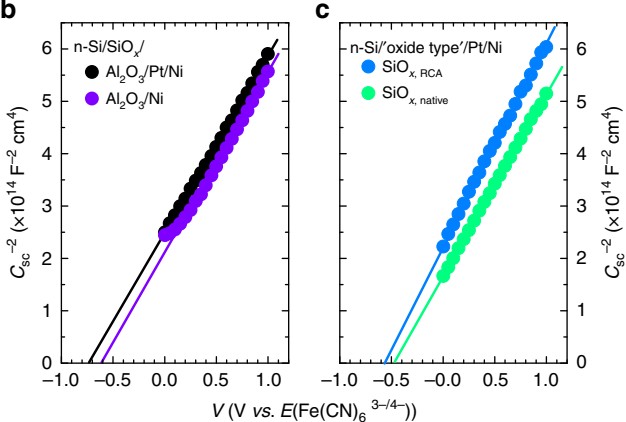

**Figure 7 | Cyclic voltammetries of MIS photoanodes with various interfacial layers and metal overlayers and their corresponding Mott–Schottky plots.** (**a**) Representative $j$–$V$ behaviour of the n-Si/$SiO_{x,RCA}$/$Al_2O_3$/Pt (2 nm)/Ni (4 nm) and n-Si/$SiO_{x,RCA}$/$Al_2O_3$/Ni (6 nm). For comparison, the samples without $Al_2O_3$ are also shown; with the $SiO_{x,RCA}$ and $SiO_{x,native}$. (**b**) Mott–Schottky plots of n-Si/$SiO_{x,RCA}$/$Al_2O_3$/Pt (2 nm)/Ni (4 nm) and n-Si/$SiO_{x,RCA}$/$Al_2O_3$/Ni (6 nm). (**c**) Mott–Schottky plots of n-Si/$SiO_{x,RCA}$/Pt/Ni and n-Si/$SiO_{x,native}$/Pt/Ni. All the plots shown are samples measured after 18 h of ageing treatment in 1 M KOH solution.

passivation by the chemically grown $SiO_{x,RCA}$ and $Al_2O_3$ stack combination.

The stability of the Ni protection layer was evaluated by a chronoamperometry test at a fixed applied potential of 1.7 V versus RHE in 1 M KOH electrolyte under simulated solar illumination, and is shown in Fig. 8a. The initial photocurrent density was 28 mA cm$^{-2}$ and periodically dropped due to the unavoidable bubble formation during the measurement. Gas bubbles were constantly generated and detached on the electrode surface, causing fluctuations in the observed photocurrent. The 4 nm Ni (or Ni/NiO$_x$/Ni(OH)$_2$ upon transformation) surface film was able to protect the underlying photoanode under illumination and at a constant applied voltage without any noticeable decay of photocurrent for more than 200 h in 1 M KOH solution. Linear sweep voltammetries were collected periodically during the chronoamperometry test (Fig. 8b) and showed no anodic shift in the $j$–$V$ characteristics during 200 h of operation. Without Pt and Ni, the $Al_2O_3$ layer was not stable in KOH solution (Supplementary Fig. 7), and thus could potentially delaminate the metal overlayers, which would subsequently cause a rapid degradation of performance. The constant observed current during the chronoamperometry measurement indicated that the $Al_2O_3$ underneath was not adversely affected by the strongly corrosive electrolyte, and demonstrates the

effectiveness of the ultrathin Ni film for electrode protection against corrosion.

## Discussion

This work clearly demonstrates the use of an MIS Si photoelectrode for the water-oxidation half reaction, and therefore does not satisfy the thermodynamic requirement for the overall water-splitting reaction. To successfully realize a spontaneous water splitting, the photoanode should be combined with larger band gap photoelectrode in a series or tandem arrangement. As a result, a considerable fraction of light would be absorbed by the top junction, and therefore, our device would ideally produce less than the reported photocurrent. However, the narrow band gap of Si and the high photovoltage as well as the high photocurrent output of our electrode could potentially simplify the design of a highly efficient PEC device for solar water splitting.

In summary we have successfully fabricated an MIS photoanode consisting of n-Si/$SiO_{x,RCA}$/$Al_2O_3$/Pt/Ni and demonstrated a high photovoltage and stability for solar water oxidation. The high photovoltage was achieved by engineering the semiconductor–insulator and insulator–metal interfaces using a thin $Al_2O_3$ dielectric layer to unpin the Si Fermi level and using a high work function metal Pt as well as an active water oxidation catalyst Ni. The incidental oxidation of Ni in 1 M KOH electrolyte resulted in an increase in its work function that subsequently improved the effective barrier height of the composite system as well as the degree of band bending near the Si surface, thereby increasing the photovoltage with a value close to the conventional np$^+$ Si buried junction. Additionally, we have shown the role of the interfacial oxide layers on the pinning and depinning of the Fermi level, which was partially responsible for the shift of onset potential, and thus the observed photovoltage. Finally we have demonstrated a simple yet effective strategy to achieve a high efficiency MIS photoanode that was chemically stable for more than 200 h in a highly corrosive environment.

## Methods

**Chemicals.** All chemicals were used as received: Potassium hydroxide pellets (KOH, Alfa Aesar, 85%), hydrogen peroxide (H$_2$O$_2$, 30% (w/w) in H$_2$O, contains stabilizer, Sigma-Aldrich), sulfuric acid (H$_2$SO$_4$, 99.999%, Sigma-Aldrich), hydrofluoric acid (HF, ACS reagent 48%, Sigma-Aldrich), hydrochloric acid (HCl, reagent grade, 37%, Sigma-Aldrich), potassium hexacyanoferrate(II) trihydrate (K$_4$Fe(CN)$_6$. 3H$_2$O, ≥99% puriss. p.a., ACS reagent, Sigma-Aldrich), potassium hexacyanoferrate(III) (K$_3$Fe(CN)$_6$, ≥99% puriss. p.a., ACS reagent, Sigma-Aldrich). Water with resistivity 18.2 MΩ cm from Milli-Q integral ultrapure water (Merck Millipore).

**Preparation of substrates.** Phosphorus-doped (n-type, (100)-oriented, single-side polished, resistivity 0.1–0.3 Ω cm, 525 μm) and degenerately boron-doped (p$^+$-type, (100)-oriented, singe-side polished, resistivity <0.005 Ω cm) Si wafers were purchased from Si-Mat. The n-type Si wafers were first cleaned in a piranha solution containing a mixture of H$_2$SO$_4$ and H$_2$O$_2$ (3:1 volume ratio) at 120 °C for 20 min to remove organic contaminants. The n-type Si wafers were then dipped into a buffered HF etchant (2%) for 2 min at room temperature to strip the native oxide on the Si wafer surface. Next, the Si wafers were immersed in a RCA SC-2 solution consisting of H$_2$O, HCl and H$_2$O$_2$ (5:1:1 by volume ratio) at 75 °C for 10 min to regrow the oxide layer ($SiO_{x,RCA}$). Finally, the Si wafers were rinsed using deionized water and dried using N$_2$ gas. The same procedure was applied on the p$^+$-type Si wafers.

**Atomic-layer deposition of aluminium oxides.** ALD of aluminium oxides (Al$_2$O$_3$) was conducted in a home-built thermal ALD system (developed at AMOLF) at 250 °C at a base pressure of 0.01–0.05 mbar. The ALD cycle consisted of a 10 ms pulse of H$_2$O, a 18 s N$_2$ purge, a 10 ms pulse of trimethylaluminium, and another 18 s N$_2$ purge to complete the cycle. The deposition rate was approximately 1.25 Å per cycle. The eight ALD cycles were used to deposit 1 nm thick Al$_2$O$_3$ on the n-type Si/$SiO_{x,RCA}$ susbtrate. The thickness of the Al$_2$O$_3$ was estimated by ellipsometer (J.A. Wollam) using dielectric models of Al$_2$O$_3$ and Si native oxide on a Si substrate.

**Sputter-deposition of metals.** Platinum (Pt) was deposited using Prevac radio frequency (rf) magnetron sputtering from a Pt target (Mateck, 99.95%, 2 inch diameter, 5 mm thickness). The Ar flow was kept at 15 sccm and the working

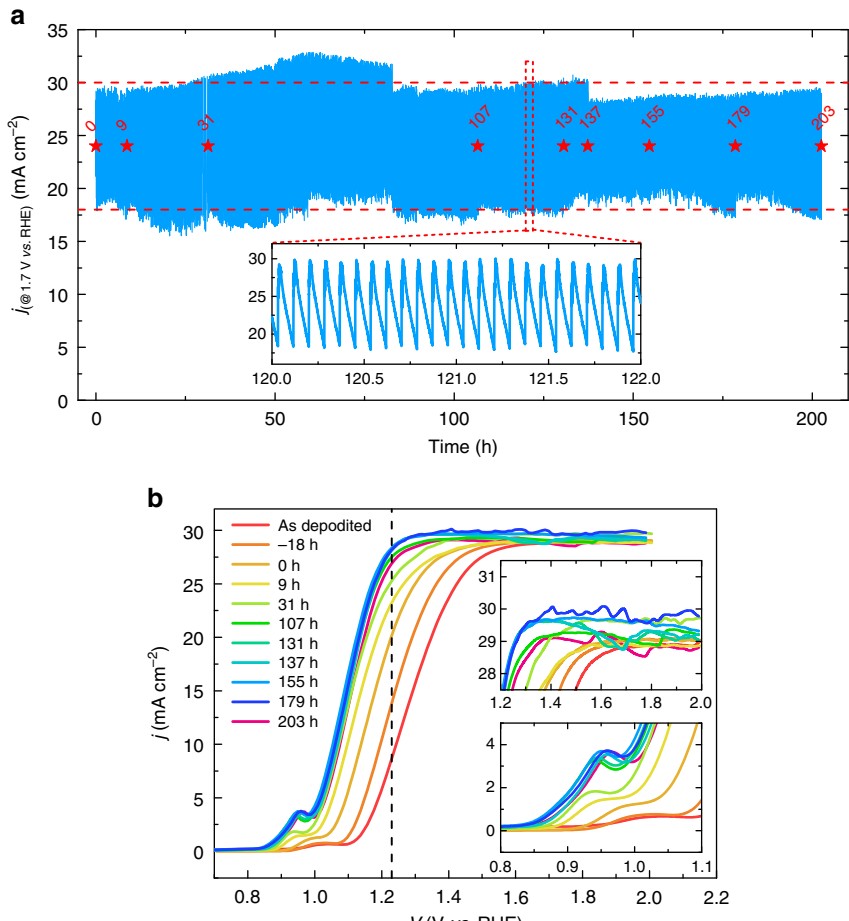

**Figure 8 | Stability of the photoanode.** (**a**) Chronoamperometic current versus time ($j$–$t$) curve of the n-Si/SiO$_{x,RCA}$/Al$_2$O$_3$/Pt/Ni photoanode measured at a constant applied voltage of 1.7 versus RHE in 1M KOH solution under simulated solar illumination. The inset shows current fluctuations due to bubble formation during the measurement. (**b**) Representative $j - V$ behaviour of the n-Si/SiO$_{x,RCA}$/Al$_2$O$_3$/Pt/Ni photoanode in contact with 1M KOH under simulated solar illumination collected periodically during 200 h of operation. Insets show the magnification of the evolution of photocurrent (top) and Ni redox waves (bottom).

pressure was held at 3 µbar. The rf power was kept at 25 W and the deposition rate was approximately, 0.138 Å s$^{-1}$. The deposition time was 144 s, and the deposited Pt film was 2 nm. Nickel (Ni) was deposited in the same Prevac sputter chamber from a pure Ni target (Mateck, 99.95%, 2 inch diameter). The Ar flow was maintained at 15 sccm and the working pressure was 3 µbar. The rf power was 100 W and the deposition rate was approximately 0.2 Å s$^{-1}$. The Ni was sputter-deposited for 200 s, resulting a 4-nm thick of Ni film.

**Preparation of electrodes.** The back sides of the n-Si samples were scratched using sand paper to remove the oxide layer, followed by cleaning the residue using ethanol. Next, the ohmic back contacts were formed by rubbing the back side surfaces of the Si samples with a Ga-In alloy (75.5:24.5 wt%, 99.9% metal basis, Alfa Aesar). The ohmic back contact of the p$^+$-Si sample was formed by depositing Pt film using sputtering.

**(Photo)electrochemical measurements.** PEC measurement of the photoanode was conducted in a three-electrode configuration in 1 M KOH electrolyte solution under simulated AM1.5 solar irradiation (100 mW cm$^{-2}$) using a Newport Sol3A Class AAA solar simulator (type 94023A-SR3) with 450 W xenon short arc lamp. A mercury/mercury oxide (Hg/HgO in 1 M KOH, Hach-Lange) electrode was used as the reference electrode, and Ni coil was used as the counter electrode. The Hg/HgO electrode had a potential of 0.9281 V versus the RHE and was calibrated using a silver/silver chloride (Ag/AgCl, in saturated KCl, Hach-Lange). The exposed area of the working electrode was 0.2826 cm$^2$. During the measurement, the electrolyte was continuously stirred using a magnetic stir bar. Cyclic voltammetry, electrochemical open-circuit, EIS and chronoamperometry measurements were performed using a potentiostat PARSTAT MC (Princeton Applied Research, AMETEK). The cyclic voltammetry data were recorded at a constant scan rate of 50 mV s$^{-1}$ and the chronoamperometric data were collected at constant potential of 1.7 V versus RHE.

**Electrochemical impedance spectroscopy.** EIS of the photoanode was performed in 50 mM K$_3$Fe(CN)$_6$, 350 mM K$_4$Fe(CN)$_6$ and 1 M KCl in a three electrode measurement using a Pt wire placed in a fritted glass tube as the reference electrode and a Pt coil as the counter electrode[9,10]. The experimental setup was kept in the dark during the measurement.

**X-ray photoelectron spectroscopy.** XPS experiment was performed using the Thermo Scientific K-alpha apparatus, equipped with an Al K-alpha X-ray source and a flood gun. Parameters used for the characterization were: spot size of 400 µm, pass energy of 50 eV, energy step size of 0.1 eV, dwell time of 50 ms and ten scans in the vicinity of the binding energy of the investigated elements. For depth profiling experiment, a careful ion etching procedure was conducted using low ion energy beam of 500 eV.

**Data availability.** Data supporting the findings of this study are available within the article and the Supplementary Information file. All other relevant data are available on reasonable request from the authors.

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

## Acknowledgements

The authors would like to thank Herman Schreuders and Joost Middelkoop from the MECS group of TU Delft for technical support, Dr Lai-Hung Lai from Center for Nanophotonics, AMOLF for fruitful discussion, and Paula Perez-Rodriguez from the PVMD group of TU Delft for technical assistance. This work is part of the research programme of the Netherlands Organization for Scientific Research (NWO) and is financed by the BioSolar Cells open innovation consortium (W.A.S., I.A.D. and B.J.T.), supported by the Dutch Ministry of Economic Affairs, Agriculture and Innovation. E.C.G. and G.W.P.A. acknowledge support from an ERC starting grant 'Nano-EnabledPV', grant number 337328.

## Author contributions

I.A.D. and W.A.S. developed conceptual idea, designed the experiments and wrote the manuscript. I.A.D., B.J.T. and G.W.P.A. performed the experiments. W.A.S. and E.C.G. interpreted data and supervised the work. I.A.D., W.A.S. and E.C.G. finalized the paper. All authors participated in discussions.

## Additional information

**Competing interests:** The authors declare no competing financial interests.

