## [Peer review file · Nature Communications]

Reviewers' comments:

Reviewer #1 (Remarks to the Author):

This is an outstanding paper that is well written and well conceived. I actually enjoyed reviewing it. The authors engineer the interfacial energetics of a photoanode by using two metals to decouple the functionalities of the Schottky contact and the water oxidation catalyst. They further improve the photovoltage by carefully designing an SiO_x and Al₂O₃ tunnel layer. My only criticism of the paper is that the stability study as done by poisoning the electrode at a potential of 1.8 V, well into the plateau region. I suspect that the photovoltage and fill factor may have degraded seriously after the 150 hour irradiation, but this would not be seen in this plot. This could be probed by simply running iV curves such as shown in Fig. 2a before and after the stability study. The other way to look at this would be to do the stability study poised at the maximum power point. If the photoelectrode is still stable under those conditions, this is a truly outstanding result.

Reviewer #2 (Remarks to the Author):

In this manuscript, the authors try to demonstrate a stable photoanode that is capable of achieving a high photovoltage for solar water oxidation by engineering the interfacial energetics of metal-insulator-semiconductor junctions. The experiments and discussions of this work are elaborated; however, the novelty of this study is not attractive enough for the readers of Nat Comm. Therefore, I cannot recommend this paper to be published. Below are some detailed comments that may help to improve the work:

1. According to the Mott-Schottky plot in Figure 4, the flat band potentials of the fresh and the aged samples should be -0.6 V and -0.73 V vs Fe(CN)₆^{3-/4-}, respectively. This error will seriously mislead the understanding of the mechanisms.
2. Obviously, Ni layer of the photoanode would suffer from corrosion in a highly corrosive electrolyte at pH 14, despite the introduction of bi-metallic layers or increasing the thickness. Then I am wondering whether the n-Si/SiO_x,RCA/Al₂O₃/Pt/NiO_x photoanode can play the same role as the structure reported in this work.
3. As the authors point out, the thickness of the Ni layer is one of the most important parameters for enhanced water oxidation activity in alkaline solution. But this paper lacks the comparison of Ni layer with different thickness.
4. The introduction of Pt reported in this study is innovative. However, Pt is known as a precious metal; whether it can be replaced by other common metal?

Reviewer #3 (Remarks to the Author):

The manuscript discusses a silicon MIS device used as a photoanode for oxygen evolution from water.

The discussion of the issues with a Schottky junction and how the solid-state junction is created and used was well-done. While this is well-known in the solid-state world, this is not well understood in the PEC water splitting community.

The manuscript is well-written, the figures are appropriate and the results are outstanding. New results include using the Pt/Ni bi-metallic layer to improve the Schottky junction and the aging effect that improves the photovoltage.

Some comments:

On page 2, paragraph starting "Herein we demonstrate...", last sentence, they state "...in a strong pH electrolyte...". What's the analytical understanding of "strong pH"? They should state exactly what the solution is, namely "strong base".

I like to suggest that they emphasize the fact that the current does not have to flow laterally in this immersed device. Because the reaction occurs everywhere at the surface of the film, the current flows perpendicular to the film. Thus the metal film can be much thinner than a solid-state device and additionally the Fermi level in the metal (the Ni/NiOx) can float to adjust for the current density and the kinetic of the reaction. Band-edge overlap is not an issue.

I understand saturated photocurrent ok, but I don't understand their logic in reporting a "short-circuit current". They need to think a little deeper into how this electrode would function in an actual water splitting device. If one looks at their figure 1b, one can see that the valance band is driving the oxygen evolution half reaction, but the Fermi level is working to drive the other half reaction, hydrogen evolution. So, under operation (true short-circuit) the Fermi level will be somewhere in between the oxygen half reaction and the hydrogen half reaction, depending on the photocathode used in this configuration.

For a two photoelectrode system, which is the possible applications for this photoelectrode, under true short circuit conditions the Fermi level will be the same for both photoelectrodes, and it will vary depending on the band edge position of the two semiconductors and the kinetics of the two half reactions. To pick any three-electrode bias for the Fermi level and call that short circuit without knowledge of the other half-reaction conditions is completely meaningless, especially if the Fermi level they choose would never be seen in a working device. I think reporting on saturated photocurrent is sufficient. However, a caveat here is that if this electrode would be combined with another (larger gap) electrode (a photocathode) for water splitting, then in a tandem arrangement, some of the light would be used by the top junction and then the saturated photocurrent would be less, ideally just half the current reported here.

I think this is appropriate for Nature Communications, the stabilization of these high efficiency semiconductors for water splitting will certainly go a long way to making the photoelectrochemical splitting of water viable.

Reviewer #1 (Remarks to the Author):

This is an **outstanding paper that is well written and well conceived**. I actually enjoyed reviewing it. The authors engineer the interfacial energetics of a photoanode by using two metals to decouple the functionalities of the Schottky contact and the water oxidation catalyst. They further improve the photovoltage by carefully designing an SiO_x and Al₂O₃ tunnel layer. My only criticism of the paper is that the *stability study as done by poisoning the electrode at a potential of 1.8 V*, well into the plateau region. I suspect that the photovoltage and fill factor may have degraded seriously after the 150 hour irradiation, but this would not be seen in this plot. This could be probed by simply running iV curves such as shown in Fig. 2a before and after the stability study. The other way to look at this would be to do the stability study poised at the maximum power point. If the photoelectrode is still stable under those conditions, this is a truly outstanding result.

We thank the reviewer for this comment and we believe this is an important proof of the stability of our device that we should clarify. We have repeated the illuminated chronoamperometry measurement at a fixed applied potential of 1.7 V *versus* RHE in contact with 1 M KOH, and collected the cyclic voltammetry periodically during 200 hours of operation. We observed that the photocurrent onset potentials *improved* during the prolonged stability test (due to the oxidation/activation of Ni) and showed no noticeable decrease in both onset potential and fill factor over time. This figure can be found in Figure 8b, page 7:

Reviewer #2 (Remarks to the Author):

In this manuscript, the authors try to demonstrate a stable photoanode that is capable of achieving a high photovoltage for solar water oxidation by engineering the interfacial energetics of metal-insulator-semiconductor junctions. The experiments and discussions of this work are elaborated; however, the novelty of this study is not attractive enough for the readers of Nat Comm. Therefore, I cannot recommend this paper to be published. Below are some detailed comments that may help to improve the work:

We thank the reviewer for their assessment of our manuscript. Before addressing the specific comments below, we would like to address the previous statement that ‘the novelty of this study is not attractive enough for the readers of Nat. Comm.’. In our work, we have demonstrated a novel approach to engineering the interfaces of MIS structures, that to date, has not been reported previously. In fact, both reviewers 1 and 3 recognize the potential impact of our work in this design of interfaces. Furthermore, many papers have been published in Nat. Mat. and Nat. Comm. (among others) on this topic,¹⁻⁵ showing that MIS structures for PEC water splitting applications are of sufficiently high interest to the readership of Nature journals. Finally, this reviewer later comments (comment 4), that the use of Pt is ‘innovative’, so it seems they actually do agree that we have made significantly novel steps in our work.

1. According to the Mott-Schottky plot in Figure 4, the flat band potentials of the fresh and the aged samples should be -0.6 V and -0.73 V vs $\text{Fe}(\text{CN})_6^{3-/4-}$, respectively. This error will seriously mislead the understanding of the mechanisms.

We thank the reviewer for pointing out our error here. In fact, in the manuscript, we have the same numbers (0.6V and 0.73V), however, we forgot to put the minus sign in front of both numbers. We have thus inserted the minus sign to show that these numbers are negative. However, since this was a mistake of a minus sign, our interpretation is still the same and does not change at all our understanding of the system and the mechanism. Therefore, changing the minus sign does not change our discussion or analysis of the phenomenon.

2. Obviously, Ni layer of the photoanode would suffer from corrosion in a highly corrosive electrolyte at pH 14, despite the introduction of bi-metallic layers or increasing the thickness. Then I am wondering whether the n-Si/SiO_x/RCA/Al₂O₃/Pt/NiO_x photoanode can play the same role as the structure reported in this work.

We have performed experiments using this structure by depositing NiO directly onto n-Si/SiO_x/Al₂O₃/Pt. The NiO was directly sputtered on the sample from a NiO ceramic target. Depending on the preparation/post-treatment of the NiO, the samples show different flat band potentials, and therefore the resulting photovoltages/photocurrent onset potentials. This is most likely related to the NiO atomic structure and/or chemical composition. It has been previously shown that the work function of NiO is sensitive to the processing conditions and environments, and can vary from 5.2 to 6.7 eV.⁶ However reporting this structure is beyond the scope of our work, and we are currently performing a separate investigation on this structure as a follow up for this work (attached figures are shown for clarity, but as they are part of a future follow up manuscript we would like them to be considered confidential). With the data that we are showing here, we are able to confirm that direct deposition of NiO onto n-Si/SiO_x/Al₂O₃/Pt can play the same role (if not better) as aging the surface metallic Ni film to NiO_x.

2. As the authors point out, the thickness of the Ni layer is one of the most important parameters for enhanced water oxidation activity in alkaline solution. But this paper lacks the comparison of Ni layer with different thickness.

The study of the effect of Ni layer thickness on a Si photoanode for water oxidation activity in alkaline solution has been shown in a well-established paper by Kenney *et. al.*⁷, and we do not feel the need to repeat the same experiment. They found that 2 nm of Ni is sufficient to effectively catalyze the Si photoanode with the same activity as thicker Ni film. However, the 2 nm Ni is not sufficient to protect the photoanode in 1 M KOH solution, and therefore thicker Ni is required to stabilize the Si photoanode but with an expense of photovoltage loss. In our work, we use a 4 nm Ni to protect our

photoanode from corrosion while still maintaining the high photovoltage using Al_2O_3 tunnel dielectric layer and Pt interfacial Schottky contact.

4. The introduction of Pt reported in this study is innovative. However, Pt is known as a precious metal; whether it can be replaced by other common metal?

We have shown in the manuscript that a non-precious material can be used as an alternative to Pt but at the expense of a lower photovoltage. This is mainly because all the high work function metals (higher than Ni work function) are precious, such as Pt, Pd, Ir. Therefore in our work, the Pt was deposited as thin as possible while still ensuring the effective charge screening effect on the semiconductor by the high work function Pt. Alternatively, one can use a thermally grown SiO_2 in place of Al_2O_3 as a surface passivation and tunnel dielectric layer. This approach has been previously demonstrated in reference 3 for a MIS photocathode consisting of p-Si/ SiO_2 /Ti/Pt. A high quality SiO_2 layer is able to unpin the Fermi level of Si from the intrinsic surface states and maintain the effective work function of a metal at its vacuum value, with a pinning factor close to unity⁸ (pinning factor 1 is for perfect unpinning and pinning factor 0 is perfect pinning. Unfortunately, growing an ultrathin SiO_2 using thermal oxidation needs special equipment and precise control that we do not have in our facilities. On the other hand, although not as efficient as SiO_2 , Al_2O_3 has similar capability to maintain the high work function of metal, and thus was used in our study as an alternative dielectric material.

Reviewer #3 (Remarks to the Author):

The manuscript discusses a silicon MIS device used as a photoanode for oxygen evolution from water.

The discussion of the issues with a Schottky junction and how the solid-state junction is created and used was well-done. While this is well-known in the solid-state world, this is not well understood in the PEC water splitting community.

The manuscript is well-written, the figures are appropriate and the results are outstanding. New results include using the Pt/Ni bi-metallic layer to improve the Schottky junction and the aging effect that improves the

photovoltage.

Some comments:

On page 2, paragraph starting “Herein we demonstrate...”, last sentence, they state “...in a strong pH electrolyte...”. What’s the analytical understanding of “strong pH”? They should state exactly what the solution is, namely “strong base”.

We agree with the reviewers comments, and change the words according to say strong base instead of strong electrolyte.

I like to suggest that they emphasize the fact that the current does not have to flow laterally in this immersed device. Because the reaction occurs everywhere at the surface of the film, the current flows perpendicular to the film. Thus the metal film can be much thinner than a solid-state device and additionally the Fermi level in the metal (the Ni/NiOx) can float to adjust for the current density and the kinetic of the reaction. Band-edge overlap is not an issue.

We thank the reviewer for the suggestions, and we have added these important points into our text in page 2, left column, line 17:

“The thin metal film allows the reaction to occur throughout the whole surface, thus enabling the photogenerated current to flow perpendicular to the metal film. Additionally, the complete coverage of the metal film results in a buried Schottky junction that isolates the internal electric field near the semiconductor surface and allows the Fermi level of the metal to float and adjust with the current density and kinetics of the reaction.”

I understand saturated photocurrent ok, but I don’t understand their logic in reporting a “short-circuit current”. They need to think a little deeper into how this electrode would function in an actual water splitting device. If one looks at their figure 1b, one can see that the valance band is driving the oxygen evolution half reaction, but the Fermi level is working to drive the other half reaction, hydrogen evolution. So, under operation (true short-circuit) the Fermi level will be somewhere in between the oxygen half reaction and the hydrogen half reaction, depending on the photocathode used in this configuration. For a two photoelectrode system, which is the possible applications for this photoelectrode, under true short circuit conditions the

Fermi level will be the same for both photoelectrodes, and it will vary depending on the band edge position of the two semiconductors and the kinetics of the two half reactions. To pick any three-electrode bias for the Fermi level and call that short circuit without knowledge of the other half-reaction conditions is completely meaningless, especially if the Fermi level they choose would never be seen in a working device. I think reporting on saturated photocurrent is sufficient.

We thank the reviewer for this comment. The terminology “equivalent illuminated short-circuit current” that we used in the manuscript is to describe the situation when the quasi-Fermi level of electron and hole are the same position, *i.e.*, flat. As an illustration, under illumination and under potential sweep, the electron Fermi level ($E_{F,n}$) equals to the applied bias, and the hole Fermi level ($E_{F,h}$) is at the same position as the Fermi level of the catalyst (E_{cat}). When the E_{cat} has reached its current onset potential (for water oxidation), the $E_{F,n}$ will approach the $E_{F,h}$ (note that $E_{F,h} = E_{cat}$) as in the reverse bias voltage in the solid-state PV measurement. This occurs because the current that flows from the catalyst to the electrolyte should be equal as the current that is generated in the Si. Because the E_{cat} will adjust with the photogenerated current output from Si, therefore we can consider the “equivalent short circuit current” as the current at which the photocurrent of the Si photoanode (n-Si/SiO_x/Al₂O₃/Pt/Ni) intersects with the current of the catalyst (p⁺Si/Ni).

However, we agree that the short-circuit terminology can be misleading and does not reflect the actual situation in a photoelectrochemical system consisting two electrodes as described above by the reviewer, and thus reporting the saturated photocurrent should be more relevant, as suggested by the reviewer. We decided to remove the related text from the manuscript.

However, a caveat here is that if this electrode would be combined with another (larger gap) electrode (a photocathode) for water splitting, then in a tandem arrangement, some of the light would be used by the top junction and then the saturated photocurrent would be less, ideally just half the current reported here.

We agree with the reviewer to clarify that the operation of our photoelectrode in the manuscript is only for the half reaction of water oxidation, and therefore should be combined with larger band gap electrodes to satisfy the thermodynamic potential for the bias free overall water

splitting reactions, and the photocurrent would ideally be less than reported current in the text. We have added an extra paragraph to clarify this in page 7, right column, line 68:

“This work clearly demonstrates the use of a MIS Si photo-electrode for the water-oxidation half reaction, and therefore does not satisfy the thermodynamic requirement for the over-all water splitting reaction. To successfully realize a spontaneous water splitting system, the photoanode should be combined with larger band gap photoelectrode in a series or tandem arrangement. As a result, a considerable fraction of light would be absorbed by the top junction, and therefore, our device would ideally produce less than the reported photocurrent. However, the narrow band gap of Si and the high photo-voltage as well as the high photocurrent output of our electrode could potentially simplify the realization of a highly efficient photoelectrochemical device for water splitting.”

I think this is appropriate for Nature Communications, the stabilization of these high efficiency semiconductors for water splitting will certainly go a long way to making the photoelectrochemical splitting of water viable.

We thank the reviewer for this kind comment about the novelty and impact of our manuscript, and agree that the key points of stability and interfacial engineering are very important for the PEC water splitting community.

1. Chen, Y. W. *et al.* Atomic layer-deposited tunnel oxide stabilizes silicon photoanodes for water oxidation. *Nat. Mater.* **10**, 539–44 (2011).
2. Scheuermann, A. G. *et al.* Design principles for maximizing photovoltage in metal-oxide-protected water-splitting photoanodes. *Nat. Mater.* 1–8 (2015). doi:10.1038/nmat4451
3. Esposito, D. V, Levin, I., Moffat, T. P. & Talin, a A. H₂ evolution at Si-based metal-insulator-semiconductor photoelectrodes enhanced by inversion channel charge collection and H spillover. *Nat. Mater.* **12**, 562–8 (2013).
4. Hill, J. C., Landers, A. T. & Switzer, J. A. An electrodeposited inhomogeneous metal–insulator–semiconductor junction for efficient photoelectrochemical water oxidation. *Nat. Mater.* **14**, 6751–6755 (2015).
5. Ji, L. *et al.* A silicon-based photocathode for water reduction with an epitaxial SrTiO₃ protection layer and a nanostructured catalyst. *Nat. Nanotechnol.* **10**, 84–90 (2014).
6. Greiner, M. T., Helander, M. G., Wang, Z.-B., Tang, W.-M. & Lu, Z.-H. Effects of Processing Conditions on the Work Function and Energy-Level Alignment of NiO Thin Films. *J. Phys. Chem. C* **114**, 19777–19781 (2010).

7. Kenney, M. J. *et al.* High-performance silicon photoanodes passivated with ultrathin nickel films for water oxidation. *Science* **342**, 836–40 (2013).
8. Yeo, Y. C., King, T. J. & Hu, C. Metal-dielectric band alignment and its implications for metal gate complementary metal-oxide-semiconductor technology. *J. Appl. Phys.* **92**, 7266–7271 (2002).

REVIEWERS' COMMENTS:

Reviewer #1 (Remarks to the Author):

The authors have addressed my concerns. They have improved an already excellent manuscript. In my opinion, the paper is ready for publication with no additional changes.

Reviewer #2 (Remarks to the Author):

We notice that the authors have addressed some of our comments. Engineering the interfaces of MIS structures is indeed very important, but not every job can meet the requirements of Nat Comm. In this paper, there are some serious inconsistencies even in the revised manuscript. I don't think these mistakes are acceptable in a rigorous scientific paper. The following are some examples:

1. Page 6 line 22, the E_{fb} of -0.7 V versus $Fe(CN)_6^{3-}/4-$ of the fresh sample vs page 4 line 68, -0.6 V;
2. Page 6 line 23-24, work function of Pt/Ni metal bilayers of 4.9 eV vs Figure 6, 4.82 eV
3. Page 4 line 70-71: C_{sc} of both samples was $3.35 \pm 0.02 \times 10^{14} \text{ F cm}^{-2}$, corresponding to a donor density (ND) of $3.54 \times 10^{16} \text{ cm}^{-3}$ vs line 142-143 in Supplementary information, Mott-Schottky plot in the main text ($1.6 \times 10^{14} \text{ F cm}^{-2}$) and the above equations, the ND was calculated to be $3.35 \times 10^{16} \text{ cm}^{-3}$ and the V_n was calculated to be 0.17 eV.

In addition, considering that E_{fb} changes with the reference electrode, the absolute value of Schottky barrier in Figure 6 and Supplementary note 2 is worth further consideration.

Reviewer #3 (Remarks to the Author):

I think that the author's responses are appropriate and what they have done certainly improved the manuscript.

I'm looking forward to seeing it in print.

Reviewer #1 (Remarks to the Author):

The authors have addressed my concerns. They have improved an already excellent manuscript. In my opinion, the paper is ready for publication with no additional changes.

We are very grateful for all the comments and time from the reviewers, and that in addressing their concerns we have made the manuscript even stronger.

Reviewer #2 (Remarks to the Author):

We notice that the authors have addressed some of our comments. Engineering the interfaces of MIS structures is indeed very important, but not every job can meet the requirements of Nat Comm. In this paper, there are some serious inconsistencies even in the revised manuscript. I don't think these mistakes are acceptable in a rigorous scientific paper. The following are some examples:

1. Page 6 line 22, the E_{fb} of -0.7 V versus $Fe(CN)_6^{3-/4-}$ of the fresh sample vs page 4 line 68, -0.6 V;
2. Page 6 line 23-24, work function of Pt/Ni metal bilayers of 4.9 eV vs Figure 6, 4.82 eV
3. Page 4 line 70-71: C_{sc-2} of both samples was $3.35 \pm 0.02 \times 10^{14} F2cm^{-4}V^{-1}$, corresponding to a donor density (ND) of $3.54 \times 10^{16} cm^{-3}$ vs line 142-143 in Supplementary information, Mott-Schottky plot in the main text ($1.6 \times 10^{14} F2cm^{-4}V^{-1}$) and the above equations, the ND was calculated to be $3.35 \times 10^{16} cm^{-3}$ and the V_n was calculated to be 0.17 eV.

In addition, considering that E_{fb} changes with the reference electrode, the absolute value of Schottky barrier in Figure 6 and Supplementary note 2 is worth further consideration.

We have corrected the above typos in the revised manuscript.

With regards to the comment "...considering the E_{fb} changes with the reference electrode, the absolute value of the Schottky barrier in Figure 6 and Supplementary note 2 is worth further consideration.":

The flat band potential (E_{FB}) was measured using electrochemical impedance spectroscopy (EIS) with a Pt wire in a fritted glass tube containing $Fe(CN)_6^{3-/4-}$ redox couple as the reference electrode. Since the Pt wire will come into equilibrium with the $Fe(CN)_6^{3-/4-}$, our reference would be the redox potential of $Fe(CN)_6^{3-/4-}$ itself. The measured equilibrium potential or the open-circuit potential of the working electrode was nearly 0 V *versus* Pt reference electrode, therefore 0 V *versus* $E(Fe(CN)_6^{3-/4-})$. In our solution, the only Faradaic process involves charge transfer of $Fe(CN)_6^{3-/4-}$, so any measured potential on the working electrode *versus* the $E(Fe(CN)_6^{3-/4-})$ can be taken as an absolute value, and should not affect the calculated barrier height. In other words, we can simply remove the "*versus* $E(Fe(CN)_6^{3-/4-})$ " from the unit (*i.e.*, barrier height = 0.77 V, instead of 0.77 V *versus* $E(Fe(CN)_6^{3-/4-})$).

It is of course a different story if we use a commercial reference electrode such as Ag/AgCl or saturated calomel electrode, because those reference electrodes have their own redox potentials and are different than the redox potential of $\text{Fe}(\text{CN})_6^{3-/4-}$.

In addition, we have compared this technique with a solid-state measurement of our MIS device in the dry condition using a thicker metal (*i.e.*, back contact is connected to the front contact and no liquid solution involved during EIS measurement), and both the impedance and Mott-Schottky results are the same. Therefore, although the reviewer mentions that our measurement/description of our E_{FB} is worth 'further consideration', we are confident in our results and analysis that our reported values are still valid, and thus this point of the referee is not addressed in our new version.

Reviewer #3 (Remarks to the Author):

I think that the author's responses are appropriate and what they have done certainly improved the manuscript.

I'm looking forward to seeing it in print.

We are very grateful for all the comments and time from the reviewers, and that in addressing their concerns we have made the manuscript even stronger.